# Inflation Forecasts and European Asset Returns: A Regime-Switching Approach

**Nicolas Pesci *** , **Jean-Philippe Aguilar** , **Victor James** and **Fabien Rouillé**

Covéa Finance, Quantitative Research Department, 8-12 rue Boissy d'Anglas, FR-75008 Paris, France
*   Correspondence: nicolas.pesci@covea-finance.fr

**Abstract:** Considering market-based inflation expectations, we show that investors' forecasts are non-linear. We capture this non-linear behavior with a Markov-switching model that allows us to identify a regime of high uncertainty, and a regime of low uncertainty and low concern about inflation. Using a complete cross-asset panel of equity sectors, bonds, and commodities, we perform regressions in both regimes including several control variables, and show that the exposure of European assets returns to implied inflation is regime-dependent. We show that inflation-indexed government bonds and oil are the best way to get exposure to slow upward revisions of future inflation that correspond to periods of rallying inflation. We thus identify alternatives to hedge oneself against revisions in inflation forecasts when inflation is considered as a variable of interest by market participants, which, in fact, corresponds to periods of breaks in the trend of realized inflation. In particular, we provide empirical evidence that some equity sectors exhibit good inflation-hedging properties.

**Keywords:** regime switching; Markov switching; inflation; asset returns; asset allocation

## 1. Introduction

After more than a decade of low inflation in the Eurozone, the COVID-19 crisis appears to be the starting point of a new cycle of rising inflation according to forecasters (see, e.g., ECB Survey (2022)). This brutal change in the macroeconomic environment and its influence on the anticipations of market participants could have a direct impact on asset returns: inflation forecasts are, indeed, directly linked to the sales and profitability of invested companies and are often taken into account while investing in the financial markets. Anticipated inflation therefore remains a major concern for most market participants such as investors, who seek a protection against inflation in order to preserve their purchasing power, and asset-liability managers, who face commitments that are partially or fully indexed to consumer prices.

In order to give a tangible definition of anticipated inflation, it is possible to introduce the implied inflation rate, also known as breakeven inflation, which is defined as the difference between the yield of a 10-year nominal bond and the yield of a 10-year inflation-indexed bond issued from the same government. There are two main reasons for this methodological choice in the construction of the implied inflation rate. First, this rate is available at a daily frequency and, thus, enables to track adjustments to inflation expectations more precisely. We believe, by following Solnik (1983), that the link between inflation and asset returns goes through inflation expectations rather than realized inflation which is measured at a lower frequency. Second, it measures expectations formed by market participants and allows to consider how future inflation is priced by investors, contrary to surveys that often include forecasts from third parties or parties not directly participating in the market, such as economists.

In this paper, we make the hypothesis that inflation expectations formed by market participants are non-linear and follow a regime-switching process. The underlying rationale behind this regime-switching hypothesis is that there could exist periods of high uncertainty

and concerns about inflation, during which inflation expectations drive asset prices, and periods of more stable inflation expectations, during which inflation is less of a concern and, thus, is not (or almost not) priced. As a result of the study, we expect to find a non-linear relationship in inflation expectations that highlights some regime-dependent hedging properties among asset classes.

The paper is organized as follows: Section 2 provides a review of existing literature. In Section 3, we present the dataset used in the paper, the market-based inflation forecasts computed from bond yields, and justify the use of a non-linear process to model their dynamic. Then, we present the control variables used to isolate the effect of implied inflation on the returns of different European assets. Section 4 presents the methodology used to identify the regimes from time series and how we estimate the exposure of each asset to inflation expectations. Finally, Section 5 presents and discusses the results obtained from model estimations.

## 2. Literature Review

The link between stock returns and inflation, be it anticipated or not, has already been widely investigated. The seminal work of Flannery and Protopapadakis (2002) evidenced that inflation risk premium in the US stock market is time dependent and disappeared after 1977 after being highly significant during the 1968–1977 period. Chen et al. (1986), using a complete set of financial and macroeconomic variables, conclude that their three inflation indicators, namely CPI (Consumer Price Index), PPI (Producer Price Index), and M1 money supply negatively impact equity values, a conclusion that is supported by former academic studies such as Fama and Schwert (1977) and Gultekin (1983).

Other studies have found empirical evidence that equities can be a good inflation hedge when used properly, either by selecting the right sectors or the right periods. For instance, Bampinas and Panagiotidis (2016) demonstrate that inflation risk exposure is mostly time varying. Moreover, they show that the hedging ability of stocks heavily depends on their sectors of activity, with energy, materials, and consumer staples stocks, appearing as the best inflation hedge in the US stock market. Let us also mention Ang et al. (2018), who studied the inflation-hedging properties of the stock market by investigating portfolios with high betas to inflation: the authors show that, despite the poor ability of the overall stock market to hedge inflation, such portfolios are a good inflation hedge and tend to overweight oil and gas stocks as well as technology stocks. These conclusions are supported by numerous recent studies that highlight the good inflation-hedging properties of real assets such as precious metals and commodities (see for instance Brière and Signori (2012), Martellini and Milhau (2013), and Amenc et al. (2009)); indeed, they can affect stock returns of companies whose activity depends positively on the price of these assets.

Unlike equities, there is no broad consensus in the literature regarding the hedging properties of rates and credit instruments. Smirlock (1986) use market surveys and find empirical evidence that interest rates were positively impacted by unanticipated inflation in the post-1979 period, resulting in lower bond prices, but that they do not respond to anticipated inflation. More recent literature remains ambiguous about inflation-hedging properties of nominal bonds. Brière and Signori (2012) show, however, that in a stable macroeconomic environment with pro-cyclical inflation, bonds feature long term inflation-hedging properties, a conclusion also supported by Amenc et al. (2009).

As previously mentioned, part of the recent literature focus on the time-varying and non-linear hedging properties of real and financial assets (see for instance Shahzad et al. (2019), Ali et al. (2020), or Salisu et al. (2020)). The case of the inflation-hedging properties of financial assets has been investigated by Boons et al. (2020) and Ang et al. (2018). They both find that the exposure of stocks to inflation is strongly time-varying and that inflation beta switches sign over time. To our knowledge, Brière and Signori (2012) is the first paper that considers inflation-hedging properties as regime-dependent. They use a structural break test to distinguish between a volatile macroeconomic regime with counter-cyclical inflation and a regime of stable macroeconomic environment with pro-cyclical inflation. They analyze

the hedging properties of common financial assets and show that their inflation-hedging properties depend heavily on macroeconomic regimes. Using a setup with time-varying hedging properties and a distinction between cyclical and counter-cyclical stocks, Spierdijk and Umar (2015) find no evidence of hedging ability from both groups until 2008 and, then, find significant inflation-hedging properties for cyclical stocks from 2008 to 2012. These conclusions support our expectations in finding a regime-switching behavior related to inflation-hedging properties of financial assets.

Our paper contributes to the existing literature by studying European assets returns with a setting that introduces both regime-dependency and asset-dependency of inflation-hedging properties. First, we confirm the non-linear behavior of European inflation expectations and identify historical regimes using a Markov-Switching model. Even if the use of a Markov-Switching model for European implied inflation is quite new, let us mention that Orlowski and Soper (2019) use a regime-switching specification to investigate the link between US implied inflation and market risk. We show that one of the phases corresponds to a period of high uncertainty about inflation, with fast and large revisions of expectations by market participants, and coincides with a trend-reversion of realized inflation. The other phase is characterized by slowly-varying inflation expectations and coincides with trending realized inflation. Second, we study the inflation-hedging properties of a cross-asset panel under each regime and find that they are heterogeneous through regimes and assets.

## 3. Dataset

Our stock returns, accounting, and fixed-income data are retrieved from Bloomberg. The sample period data ranges from 12 June 2009 to 2 November2021 for a total of 3168 daily observations. This sample period is constrained by data availability for German inflation indexed bonds that are used to compute a proxy for the European implied inflation. Indeed, although inflation-linked bonds date back to the 1980s in the UK, the first Eurozone inflation-linked bond was issued in France in 2001. However, due to the rare issuance of such bonds, German inflation-linked securities offer the longest continuously quoted yield with limited sovereign risk in the Euro are, with data starting from 2009.

### 3.1. Implied Inflation

Implied inflation is a market-based measure of expected inflation, also referred to as breakeven inflation. We prefer implied inflation to other indicators such as CPI forecasts since implied inflation directly measures how investors' views are integrated in the market, whereas answers to polls can differ from the reality of investment decisions: indeed, the forecaster is not necessarily the investor. We note that, furthermore, such a metric has been used as a proxy for investors' inflation forecasts in previous academic studies such as Church (2019) and Cette and De Jong (2013).

Our first goal is to understand the dynamic of European implied inflation and, more specifically, to identify potential regime switches that could lead to changes in the exposure of asset returns to inflation priors. We define the *implied inflation* at date $t$ as the following spread:

$$\pi_t = \rho_t - \rho_{i_t} \tag{1}$$

where $\rho_t$ is the yield at date $t$ of a 10Y German bond and $\rho_{i_t}$ is the yield at date $t$ of a 10Y inflation-linked German bond. We favor long-term bonds to compute implied inflation following the conclusions of Church (2019), which show that the accuracy of breakeven inflation improves with the considered time horizon. Note that inflation-linked German bonds are indexed to European inflation, and therefore $\pi_t$ is actually a good proxy of European implied inflation itself. Let us also recall that, to ensure a parsimonious specification and inference, one should work with a stationary time series. For this purpose, we use the first difference of $\pi = [\pi_1, ..., \pi_T]$, which takes the form of a vector $r = [r_1, ..., r_{T-1}]$ where

$$r_t = \log(\pi_{t+1}) - \log(\pi_t) \tag{2}$$

for all $t \in [1, T-1]$.

The next step is to remove outliers (notably induced by the roll method used in most generic fixed income indices) from the dataset, in order to limit their impact on the estimated parameters. To that extent, we consider an AR(1) model and we compute the Cook's Distance (see details, e.g., in Atkinson and Riani (2000)) to remove from the dataset any point where Cook's Distance is greater than $\frac{16}{T-1}$. This threshold is greater than the commonly admitted $\frac{4}{T-1}$ threshold; this is because we want to remove the most influential data points without significantly reducing the size of the dataset. Hence, 65 observations have been removed, corresponding to a 2% deletion of the initial dataset.

Finally, we conduct several tests to investigate the non-linearity of implied inflation in the Euro area. Both Jarque–Bera and Shapiro–Wilk tests reject the null hypothesis of normality of the data. Hence, we run non-parametric tests of non-linearity to justify the use of a regime-switching model. The results are presented in Table 1. As one can see, each of the conducted tests reject the null hypothesis of linearity at a 1% threshold.

**Table 1.** Results of normality and non-linearity tests for the first differences of implied inflation.

| Test | Test Statistic | *p*-Value |
|---|---|---|
| Jarque and Bera (1980) | 42.69 | 0.00 *** |
| Shapiro and Wilk (1965) | 0.99 | 0.00 *** |
| Peña and Rodriguez (2002) | 6.28 | 0.00 *** |
| McLeod and Li (1983) | 3630 | 0.00 *** |

*** indicates significance at the 1% level.

### 3.2. Control Variables

Following Goyal and Welch (2008) and Kessler et al. (2021), we propose a set of control variables that are commonly used as predictors for both in-sample and out-of-sample purposes, and cover both equity and fixed income markets as well as ECB monetary policy; introducing these variables allow us to better isolate the impact of changes in market participants' inflation forecasts on asset returns. A detailed description of the control variables used for regressions is provided in Table 2. The market return is chosen as the return of the MSCI EMU Index, which is a free-float weighted equity index representative of the largest companies of the European Economic and Monetary Union. All accounting variables are computed at the index level. This index is rarely used as an underlying for options. Hence, we chose to use the V2X Index as a proxy for volatility. It is designed to measure the implied volatility of Euro Stoxx 50 Index options traded on Eurex. Although smaller, it is a leading blue-chip index for the Eurozone commonly used as an underlying index for derivative products with sufficient liquidity to compute a reliable measure of expected volatility.

Although our study focuses on inflation-hedging properties of European assets, the significance of some control variables can be of interest for decision makers. This is particularly true for the role of interest rates considering the current context of tightening monetary policy. The composition of our cross-asset panel is detailed in Table 3. As one can see, we chose to investigate the impact of inflation forecasts at a sector-level on stock returns. This choice is motivated by the results of Sarwar et al. (2018), who show a very heterogeneous behavior at the sector-level across the business cycles, which could be explained by different exposures to key variables.

Regarding the use of the MSCI EMU as reference index, the choice is motivated by its large number of members (to ensure a representation of all GICS sectors), its focus on Eurozone stocks, as well as the availability of financial data at the index-level. All stock indices are gross total return indices. For fixed-income products, we have selected four asset classes including inflation-linked government bonds, investment grade and high yield bonds and convertible bonds, which are often assumed to behave as a hybrid between stocks and bonds. We also add three of the most traded commodities, namely Brent, WTI, and gold.

**Table 2.** Definition of the control variables used in linear regressions.

| Symbol | Variable | Definition |
|--------|----------|------------|
| RM | Market returns | MSCI EMU Gross Total Returns |
| RF | Risk-free rate | 1-day yield equivalent to the yield to maturity of a 3-month German government bond considering continuous compounding |
| ECB | ECB Main Refinancing Rate | Rate for the Eurosystem's regular open market operation for a maturity of one week |
| y | 10Y German government bond yield | Yield to Maturity of a generic 10Y bond issued by German government |
| V2X | V2X Index | Implied volatility measure of the Euro Stoxx 50 Index |
| P/E | Price/Earnings Ratio | Price of a stock divided by Trailing 12M Earnings Per Share |
| D/P | Dividend Payout Ratio | Fraction of net income a firm pays to its shareholders in dividends |
| P/B | Price to Book Ratio | Fraction of net income a firm pays to its shareholders in dividends |
| DIVY | MSCI EMU Dividend Yield | Gross dividend per share aggregated on the last 12 months divided by the security price |
| ZSpr | Euro Corporates Z-spread | The zero-volatility spread (or z-spread) is the spread to be added to the spot curve to reprice a bond. This variable corresponds to the z-spread of the Bloomberg Euro-Aggregate : Corporates Index |

**Table 3.** Panel composition. The equity class is represented by several sector-based categories (finance, energy...), the fixed income class by four types of bonds (high yield, convertibles ...) and the commodity class by three different tangible assets (gold, WTI, Brent).

| Asset Class | Bloomberg Ticker | Description |
|-------------|------------------|-------------|
| Consumer Discretionary | GDLUCDIS Index | MSCI EMU/CON DIS |
| Information Technology | GDLUIT Index | MSCI EMU/INF TECH |
| Industrials | GDLUIND Index | MSCI EMU/INDUSTRL |
| Finances | GDLUFNCL Index | MSCI EMU/FINANCE |
| Consumer Staples | GDLUCSTA Index | MSCI EMU/CONS STPL |
| Healthcare | GDLUHC Index | MSCI EMU/HLTH CARE |
| Materials | GDLUMAT Index | MSCI EMU/MATERIALS |
| Utilities | GDLUUTI Index | MSCI EMU/UTILITY |
| Communication Services | GDLUTEL Index | MSCI EMU/COMM SVC |
| Energy | GDLUENR Index | MSCI EMU/ENERGY |
| Real Estate | GDLURLCL Index | MSCI EMU/REAL ESTATE |
| Investment Grade Credit | ER00 Index | ICE BofA Euro Corporate Index |
| High Yield Credit | HE00 Index | ICE BofA Euro High Yield Index |
| Inflation-linked Government Bonds | EG0I Index | ICE BofA Euro Inflation-Linked Index |
| Convertible Bonds | EZCIEZCI Index | Exane Eurozone Convertible Bonds Index |
| Gold | XAU Curncy | Gold Spot $/oz |
| WTI | CL1 Comdty | Generic 1st 'CL' Future |
| Brent | CO1 Comdty | Generic 1st 'CO' Future |

## 4. Model

Using our implied inflation data, we estimate the following MS(k)-AR(p) model :

$$r_t = \phi_{0,S_t} + \sum_{j=1}^{p} \phi_{j,S_t} r_{t-j} + \varepsilon_t, \tag{3}$$

where $S_t = \{1, ..., k\}$ is a first-order Markov chain governing the state of the process, $\varepsilon_t$ is a sequence of independent and identically distributed random variables with zero mean and variance $\sigma_{S_t}^2$, and $\phi_{j,S_t}$ is the autoregressive coefficient for lag $j$ in state $S_t$. We can note that $S_t$ is not observable and, thus, is treated as a latent variable. We also introduce the $k \times k$ transition matrix of the state variable, denoted by $\boldsymbol{P} = (p_{ij})_{i,j=1...k}$, and satisfying

$$p_{ij} = \mathbb{P}(S_t = j \,|\, S_{t-1} = i) \tag{4}$$

for all $i, j = 1, \ldots k$. Model parameters are estimated using the maximum likelihood estimator proposed in the seminal work of Hamilton (1989). This estimation procedure goes through a filtering step and a smoothing step that allow us to compute the smoothed probabilities $\mathbb{P}(S_t = j \,|\, y_T, \ldots, y_1; \theta)$ of being in state $j$ at date $t$ conditionally to our sample up to date $T$ and to the set of parameters to be estimated and denoted by $\theta$.

To our knowledge, no other work has been conducted to identify regimes of market participants' inflation forecasts in the Euro area, however, similar works may exist on realized inflation data at lower frequency (see Nalewaik (2015) for instance) and for implied inflation in the US (see Orlowski and Soper (2019)). To address this issue, we conduct specification tests to ensure robust parametrization (let us recall that we showed in Section 3 that the choice of a non-linear model is justified). We use a consistent selection criterion that penalizes complex parametrization, namely the Bayesian Information Criterion (BIC), to select the number of regimes and the number of autoregressive terms to be used in our model. It appears that the model minimizing our selection criterion includes two regimes and one lag but no intercept. This result is consistent with other studies investigating Markov switching models (with different specifications) on CPI data and finding two regimes, see for instance Song (2017) and Nalewaik (2015). In this context, we recall that one can compute the expected duration of regime $i$, denoted by $\mathbb{E}[d_i]$, using transition probabilities to assess the persistence of each regime:

$$\mathbb{E}[d_i] = \sum_{k=1}^{\infty} k \mathbb{P}(d_i = k) = \frac{1}{1 - p_{ii}}. \tag{5}$$

Once the MS-AR model has been estimated, we are able to identify the state of the process at any time $t$ with smoothed probabilities. We classify each date $t$ as being in regime $i$ if $\mathbb{P}(S_t = j \,|\, y_T, \ldots, y_1; \theta) > 0.5$. Let $\{t_{S_t}\}$ be the set of dates for which the system is in regime $S_t$. We can then split the dataset between $r_{t_1}$ and $r_{t_2}$ corresponding respectively to the set of implied inflation observations identified as being in regime 1 and regime 2. We run the following regressions in order to estimate the factor loading of each variable with a specific focus on interest rates and implied inflation:

$$R_{it_{S_t}} - RF_{t_{S_t}} = c_{i_{S_t}} + \beta_{i1_{S_t}} \Delta y_{t_{S_t}} + \beta_{i2_{S_t}} \Delta \pi_{t_{S_t}} + \beta_{i3_{S_t}} (RM_{t_{S_t}} - RF_{t_{S_t}})$$

$$+ \sum_{j=4}^{n} \beta_{ij_{S_t}} \Delta x_{j_{S_t}} + \varepsilon_{it_{S_t}} \text{ for } S_t = \{1, 2\}, \tag{6}$$

where $R_{it_{S_t}}$ is the return of asset $i$ for the $t$-th observation of regime $S_t$, $\beta_{ij_{S_t}}$ are real-valued parameters to be estimated, $x_{j_{S_t}}$ represents control variables, $c_{i_{S_t}}$ is the intercept of the regression and the $\Delta$ symbol indicates we use the change of variables. Our cross-asset panel includes 18 assets with 2 regimes, resulting in 36 regressions. In this setup, the exposure of

each asset to inflation anticipations is captured by $\beta_{ij_{S_t}}$ whose sign and p-value will be of particular interest to conclude about the link between excess returns and implied inflation.

## 5. Results

The results of the estimation procedure applied to the time series of 100 times the change in implied inflation is reported in Table 4. As one can see, all estimated parameters are significant at the 1% level. The sign of the autoregressive coefficient is the same in both regime and with close values, so the difference between the two dynamics must be found elsewhere.

**Table 4.** MS(2)-AR(1) model estimation.

| $p_{11}$ | $p_{21}$ | $\sigma_1$ | $\sigma_2$ | $\phi_{1,1}$ | $\phi_{1,2}$ | $E[d_1]$ | $E[d_2]$ |
|---|---|---|---|---|---|---|---|
| 0.969 *** | 0.046 *** | 1.204 *** | 6.670 *** | 0.220 *** | 0.200 *** | 32.748 | 21.571 |
| (0.007) | (0.011) | (0.089) | (0.443) | (0.026) | (0.029) | | |

*** indicates significance at the 1% level.

### 5.1. Implied Inflation and Volatility Regimes

We present sample statistics in Table 5 for each regime. As one can see, the mean of the implied inflation remains quite low in both regimes, but the standard deviation is significantly different between the two regimes. More precisely, with a standard deviation of 2.66% in regime 2, the volatility of inflation expectations is more than twice higher than in regime 1. Moreover, the distribution is skewed to the left under regime 1, which shows that regime 2 features large negative changes.

In order to explore this behavior a bit further, we plot in Figure 1 the log-change in implied inflation over time and shade the dates for which the smoothed probability of regime 2 is over 0.5: it turns out that shaded areas correspond to periods of high volatility of inflation expectations. This clearly suggests that inflation forecasts are subject to small changes in regime 1, reflecting the fact that even if new information is added, it is almost completely included in current inflation expectations. In other words, these changes should have already been included in asset prices by market participants, and, as a consequence, one should expect little or no impact from changes in implied inflation in regime 1. On the contrary, one can clearly observe that uncertainty around inflation in regime 2 is higher, and that exogenous shocks have a higher impact on inflation expectations, resulting in in major revisions of future inflation.

**Table 5.** Descriptive statistics of log-differences of implied inflation in each regime.

| | Regime 1 | Regime 2 |
|---|---|---|
| Mean | 0.0187% | −0.0346% |
| Standard deviation | 1.1155% | 2.6634% |
| Skewness | 0.0900 | −0.466 |

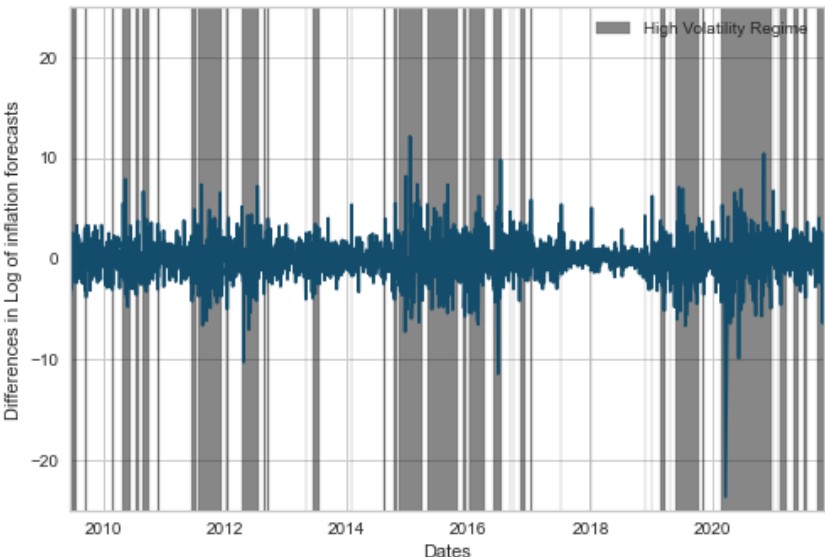

**Figure 1.** Implied inflation changes through volatility regimes.

### 5.2. Realized Inflation and Volatility Regimes

In Figure 2, one can observe that regime 2 seems to correspond to periods of breaks in the trend of realized inflation: in such situations, the most expected scenario of market participants is strongly revised, which could be reflected in asset prices. In this regime, where future inflation becomes a concern, we therefore expect European asset returns to have a significant exposure to changes in implied inflation. We note that the expected duration of each regime is quite large, which is consistent with the fact that key variables affecting inflation forecasts are released at a low frequency. Hence, switching from a regime to another is a slow process that requires the accumulation of signals to confirm the reversion of realized inflation and to form a new consistent forecast.

### 5.3. Regressions on Asset Classes

The results of the regressions for each asset class for the low and the high volatility regime are presented in Tables 6 and 7, respectively. Under the low volatility regime, we find empirical evidence that a significant exposure exists across almost all assets of our panel to changes in interest rates, with the exception of materials and communication services equity sectors as well as oil-related commodities. For equities, the impact of a positive change in interest rates is globally negative, which can be explained by both a higher discount factor used for DCF-based valuations (widely used for sectors such as information technology) and a higher cost of debt (for the most indebted sectors such as utilities and real estate for instance). As one can see in Table 7, the return exposure for these sectors to interest rates remains strong in the regime of high volatility of inflation forecasts. As expected, the full panel of fixed income assets presents a significant negative exposure to interest rates variations in both regimes, with convertible bonds showing the lowest beta, which is consistent when considering the hybrid structure of this asset.

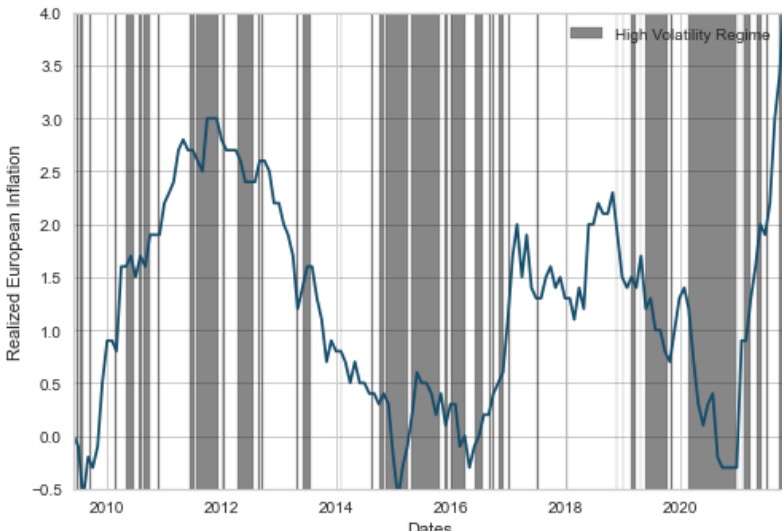

**Figure 2.** Realized inflation through volatility regimes.

Regarding the exposure of our panel to changes in implied inflation, inflation-linked government bonds and oil products present a significant and positive exposure to the variable in both regimes, making them the best hedges to rising inflation forecasts. This result is consistent with the fact that, when investors revise their inflation forecasts upward, they expect a higher coupon from these products due to the inflation indexation. However, as one can see in Tables 6 and 7, the response of inflation-linked bonds is higher in the high volatility regime, which can be counter-intuitive given that we have just seen that the low volatility regime corresponds to periods of large trends in realized inflation to which the coupon is indexed. In fact, it seems that investors quickly incorporate their views in inflation-linked bonds prices and that the rallying realized inflation is mostly priced at the end of the reverting period. This is why the coefficient associated with changes in implied inflation in regime 1 (0.442) remains relatively low when compared to the coefficient associated with changes in interest rates ($-3.862$). The positive link between Brent/WTI and implied inflation is less surprising, considering that energy prices are one of the major components of inflation.

For other (non inflation-linked) fixed income products as well as for equity sectors, the situation is slightly different. Indeed, we find evidence of a time-varying loading depending on the regime of inflation forecasts. As expected, in the regime of low uncertainty on future inflation, we find no evidence of a significant effect of implied inflation on these assets, indicating that an upward revision does not lead to any movement in equity returns in this regime. This means that, in regime 1, changes in implied inflation are not priced in equity returns by market participants, supporting the fact that the variable is not of interest for equity investors in this regime. The same conclusion applies to fixed-income products (except inflation-linked bonds), as well as to gold returns.

**Table 6.** Regression results for our panel under the low volatility regime of inflation expectations from 12 June 2009 to 02 November 2021. Error estimates are indicated in parentheses except for the F-Test, for which the *p*-value is in parentheses.

| | Intercept | y | π | ECB | V2X | RM-RF | P/E | P/B | D/P | DIVY | ZSpr | F-Test | Adj. R² |
|---|---|---|---|---|---|---|---|---|---|---|---|---|---|
| Consumer Discretionary | 0.003 (0.011) | −0.882 ** (0.351) | −0.033 (0.226) | −2.793 ** (1.227) | −0.023 * (0.013) | 1.011 *** (0.048) | 0.014 (0.019) | 3.130 (2.412) | −0.007 (0.008) | 1.091 (0.741) | 0.004 (0.009) | 753.5 (0.0) | 0.796 |
| Information Technology | 0.029 * (0.016) | −1.935 *** (0.517) | −0.187 (0.333) | 0.325 (1.809) | −0.031 (0.019) | 0.870 *** (0.071) | −0.028 (0.028) | 9.629 *** (3.557) | −0.005 (0.012) | 2.607 ** (1.093) | −0.011 (0.013) | 279.5 (0.0) | 0.591 |
| Industrials | 0.013 (0.008) | −0.945 *** (0.260) | 0.146 (0.167) | −2.274 ** (0.909) | 0.005 (0.010) | 0.997 *** (0.036) | −0.017 (0.014) | 3.373 * (1.788) | 0.005 (0.006) | −0.107 (0.549) | 0.005 (0.007) | 1349.9 (0.0) | 0.875 |
| Financials | −0.030 ** (0.013) | 3.973 *** (0.409) | 0.166 (0.263) | 0.803 (1.432) | 0.042 *** (0.015) | 1.270 *** (0.056) | 0.023 (0.022) | −6.184 ** (2.816) | −0.004 (0.009) | −1.874 ** (0.865) | −0.047 *** (0.011) | 885.8 (0.0) | 0.821 |
| Consumer Staples | 0.024 ** (0.012) | −2.194 *** (0.378) | −0.303 (0.243) | 0.304 (1.323) | −0.008 (0.014) | 0.712 *** (0.052) | 0.009 (0.020) | −0.759 (2.600) | 0.007 (0.008) | 0.221 (0.799) | 0.024 ** (0.010) | 266.5 (0.0) | 0.580 |
| Healthcare | 0.011 (0.014) | −1.350 *** (0.444) | −0.266 (0.286) | 1.911 (1.555) | −0.001 (0.017) | 0.935 *** (0.061) | −0.021 (0.024) | −1.518 (3.056) | 0.007 (0.010) | 1.451 (0.939) | 0.019 * (0.012) | 300.1 (0.0) | 0.609 |
| Materials | 0.000 (0.011) | 0.006 (0.355) | 0.139 (0.229) | −0.424 (1.244) | 0.006 (0.013) | 0.956 *** (0.049) | −0.003 (0.019) | 4.648 * (2.445) | 0.002 (0.008) | −1.875 ** (0.752) | 0.006 (0.009) | 802.1 (0.0) | 0.806 |
| Utilities | 0.008 (0.014) | −1.367 *** (0.458) | 0.048 (0.295) | 1.627 (1.603) | 0.016 (0.017) | 0.953 *** (0.063) | 0.008 (0.025) | −7.303 ** (3.150) | 0.003 (0.010) | −0.980 (0.968) | 0.006 (0.012) | 295.1 (0.0) | 0.605 |
| Communication Services | −0.013 (0.014) | −0.086 (0.439) | −0.350 (0.283) | 2.181 (1.537) | −0.009 (0.016) | 0.827 *** (0.060) | −0.018 (0.024) | −2.533 (3.021) | 0.007 (0.010) | −0.723 (0.929) | 0.004 (0.011) | 300.8 (0.0) | 0.609 |
| Energy | 0.001 (0.015) | 1.325 *** (0.478) | 0.437 (0.308) | 0.880 (1.673) | −0.031 * (0.018) | 0.978 *** (0.066) | 0.014 (0.026) | −0.508 (3.288) | −0.004 (0.011) | 0.910 (1.011) | 0.042 *** (0.012) | 361.7 (0.0) | 0.652 |
| Real Estate | −0.009 (0.018) | −4.169 *** (0.567) | 0.400 (0.365) | −2.221 (1.984) | −0.010 (0.021) | 0.813 *** (0.078) | −0.029 (0.030) | −6.910 * (3.899) | 0.004 (0.013) | −0.695 (1.198) | −0.016 (0.015) | 135.8 (0.0) | 0.412 |
| Inv. Grade Bonds | 0.012 *** (0.001) | −3.271 *** (0.039) | −0.010 (0.025) | −0.226 * (0.135) | 0.004 *** (0.001) | 0.015 *** (0.005) | −0.001 (0.002) | −0.403 (0.266) | 0.000 (0.001) | −0.130 (0.082) | −0.027 *** (0.001) | 759.3 (0.0) | 0.798 |
| High Yield Bonds | 0.025 *** (0.002) | −1.269 *** (0.078) | 0.070 (0.050) | 0.269 (0.273) | −0.001 (0.003) | 0.032 *** (0.011) | 0.001 (0.004) | −0.360 (0.536) | −0.000 (0.002) | 0.018 (0.165) | −0.084 *** (0.002) | 229.0 (0.0) | 0.542 |
| Infla. linked Gvt Bonds | 0.004 (0.005) | −3.863 *** (0.148) | 0.442 *** (0.095) | −0.246 (0.518) | −0.004 (0.006) | 0.066 *** (0.020) | −0.005 (0.008) | −2.294 ** (1.017) | 0.002 (0.003) | −0.803 ** (0.313) | −0.041 *** (0.004) | 75.8 (0.0) | 0.280 |
| Convertible Bonds | 0.008 ** (0.004) | −0.882 *** (0.113) | 0.003 (0.073) | 0.106 (0.397) | −0.005 (0.004) | 0.256 *** (0.016) | −0.005 (0.006) | 2.154 *** (0.780) | 0.000 (0.003) | 0.394 (0.240) | −0.019 *** (0.003) | 525.2 (0.0) | 0.731 |
| Gold | 0.004 (0.021) | −3.312 *** (0.670) | 0.490 (0.431) | 1.502 (2.344) | −0.005 (0.025) | 0.123 (0.092) | −0.007 (0.036) | −7.827 * (4.607) | 0.000 (0.015) | −2.312 (1.416) | 0.003 (0.017) | 3.7 (0.0) | 0.014 |
| WTI | 0.040 (0.038) | 1.764 (1.211) | 2.324 *** (0.779) | 5.113 (4.237) | 0.003 (0.045) | 0.380 ** (0.166) | 0.041 (0.065) | 3.159 (8.329) | −0.015 (0.027) | −2.574 (2.560) | 0.043 (0.031) | 18.6 (0.0) | 0.084 |
| Brent | 0.044 (0.036) | 0.656 (1.142) | 2.033 *** (0.735) | 4.678 (3.999) | −0.015 (0.043) | 0.393 ** (0.157) | 0.012 (0.061) | 0.390 (7.860) | −0.003 (0.026) | −1.898 (2.416) | 0.012 (0.030) | 17.7 (0.0) | 0.080 |

* indicates significance at the 1% (***), 5% (**), and 10% (*) levels, respectively.

**Table 7.** Regression results for our panel under the high volatility regime of inflation expectations from 12 June 2009 to 02 November 2021. Error estimates are indicated in parentheses except for the F-Test, for which the *p*-value is in parentheses.

| | Intercept | y | $\pi$ | ECB | V2X | RM-RF | P/E | P/B | D/P | DIVY | ZSpr | F-Test | Adj. R² |
|---|---|---|---|---|---|---|---|---|---|---|---|---|---|
| Consumer Discretionary | 0.039 ** (0.016) | −0.289 (0.428) | 0.099 (0.537) | −0.603 (1.006) | 0.015 (0.013) | 1.111 *** (0.068) | −0.004 (0.065) | −1.024 (3.145) | 0.002 (0.034) | 1.194 (0.844) | 0.003 (0.006) | 831.2 (0.0) | 0.872 |
| Information Technology | 0.007 (0.024) | −2.032 *** (0.634) | −0.512 (0.796) | 0.268 (1.490) | −0.047 ** (0.020) | 0.705 *** (0.100) | 0.198 ** (0.096) | 9.802 ** (4.658) | −0.096 * (0.050) | −0.715 (1.250) | 0.020 ** (0.009) | 296.4 (0.0) | 0.708 |
| Industrials | 0.005 (0.013) | −0.328 (0.334) | 0.356 (0.419) | −0.580 (0.785) | 0.024 ** (0.011) | 1.055 *** (0.053) | 0.113 ** (0.051) | −2.903 (2.454) | −0.057 ** (0.026) | −0.967 (0.658) | −0.005 (0.005) | 1432.4 (0.0) | 0.922 |
| Financials | 0.003 (0.022) | 5.054 *** (0.579) | 0.386 (0.727) | 0.134 (1.361) | 0.060 *** (0.018) | 1.453 *** (0.091) | −0.032 (0.088) | −10.715 ** (4.254) | 0.031 (0.046) | 1.003 (1.141) | −0.038 *** (0.008) | 760.1 (0.0) | 0.862 |
| Consumer Staples | 0.010 (0.018) | −3.583 *** (0.462) | −0.669 (0.580) | 0.049 (1.087) | −0.006 (0.015) | 0.638 *** (0.073) | 0.081 (0.070) | 11.793 *** (3.398) | −0.058 (0.036) | 1.887 ** (0.912) | 0.000 (0.007) | 331.0 (0.0) | 0.730 |
| Healthcare | −0.006 (0.022) | −2.202 *** (0.564) | −2.218 *** (0.708) | −0.830 (1.327) | −0.022 (0.018) | 0.686 *** (0.089) | 0.024 (0.086) | 10.462 ** (4.147) | −0.024 (0.044) | −0.280 (1.113) | 0.033 *** (0.008) | 281.0 (0.0) | 0.697 |
| Materials | 0.000 (0.016) | 0.197 (0.416) | 0.605 (0.523) | −0.186 (0.979) | 0.005 (0.013) | 1.122 *** (0.066) | −0.076 (0.063) | −5.953 * (3.060) | 0.035 (0.033) | −0.664 (0.821) | 0.013 ** (0.006) | 902.8 (0.0) | 0.881 |
| Utilities | −0.016 (0.021) | −1.181 ** (0.558) | −0.280 (0.701) | 0.550 (1.313) | −0.051 *** (0.018) | 0.844 *** (0.088) | −0.160 * (0.085) | −6.838 * (4.105) | 0.093 ** (0.044) | −3.120 *** (1.101) | 0.013 (0.008) | 347.7 (0.0) | 0.740 |
| Communication Services | −0.017 (0.020) | −0.609 (0.534) | −0.536 (0.670) | 0.439 (1.255) | −0.029 * (0.017) | 0.852 *** (0.084) | −0.161 ** (0.081) | 3.834 (3.923) | 0.075 * (0.042) | 0.710 (1.053) | 0.021 *** (0.008) | 360.9 (0.0) | 0.747 |
| Energy | 0.000 (0.029) | 0.119 (0.768) | 4.098 *** (0.964) | 1.870 (1.806) | −0.010 (0.024) | 0.776 *** (0.121) | 0.237 ** (0.117) | 12.477 ** (5.644) | −0.146 ** (0.061) | 0.522 (1.514) | −0.016 (0.011) | 272.1 (0.0) | 0.690 |
| Real Estate | −0.012 (0.027) | −4.625 *** (0.715) | 2.152 ** (0.898) | 0.763 (1.683) | 0.014 (0.023) | 0.796 *** (0.113) | −0.084 (0.109) | −3.481 (5.260) | 0.065 (0.056) | −2.905 ** (1.411) | −0.025 ** (0.010) | 188.1 (0.0) | 0.606 |
| Inv. Grade Bonds | 0.011 *** (0.002) | −3.298 *** (0.061) | 0.418 *** (0.077) | −0.325 ** (0.144) | 0.009 *** (0.002) | 0.026 *** (0.010) | −0.002 (0.009) | 0.003 (0.449) | 0.004 (0.005) | 0.080 (0.120) | −0.028 *** (0.001) | 391.4 (0.0) | 0.762 |
| High Yield Bonds | 0.024 *** (0.006) | −1.537 *** (0.144) | 0.646 *** (0.181) | −0.226 (0.339) | 0.006 (0.005) | 0.049 ** (0.023) | −0.008 (0.022) | 1.536 (1.061) | 0.009 (0.011) | 0.336 (0.285) | −0.080 *** (0.002) | 258.3 (0.0) | 0.679 |
| Infla. linked Gvt Bonds | 0.005 (0.008) | −5.173 *** (0.210) | 5.933 *** (0.264) | 0.267 (0.495) | 0.003 (0.007) | 0.115 *** (0.033) | −0.080 ** (0.032) | −1.918 (1.548) | 0.044 *** (0.017) | −0.200 (0.415) | −0.015 *** (0.003) | 108.5 (0.0) | 0.469 |
| Convertible Bonds | 0.001 (0.006) | −1.058 *** (0.150) | 0.530 *** (0.188) | −0.047 (0.352) | −0.012 ** (0.005) | 0.184 *** (0.024) | 0.034 (0.023) | 3.545 *** (1.100) | −0.020 * (0.012) | 0.091 (0.295) | −0.016 *** (0.002) | 437.5 (0.0) | 0.782 |
| Gold | 0.021 (0.029) | −4.167 *** (0.771) | 1.320 (0.968) | 0.856 (1.813) | −0.101 *** (0.024) | 0.228 * (0.122) | −0.412 *** (0.117) | −18.873 *** (5.667) | 0.217 *** (0.061) | 0.769 (1.521) | −0.015 (0.011) | 7.0 (0.0) | 0.047 |
| WTI | −0.330 (0.287) | −5.431 (7.511) | 21.520 ** (9.432) | −1.981 (17.667) | −0.452 * (0.237) | −0.207 (1.187) | 0.708 (1.142) | −9.414 (55.222) | −0.373 (0.592) | −6.770 (14.817) | −0.025 (0.108) | 2.5 (0.0) | 0.012 |
| Brent | −0.007 (0.071) | 0.899 (1.869) | 13.252 *** (2.347) | −1.204 (4.397) | −0.105 * (0.059) | −0.129 (0.296) | 0.270 (0.284) | 19.138 (13.744) | −0.211 (0.147) | −3.754 (3.688) | −0.050 * (0.027) | 23.3 (0.0) | 0.154 |

* indicates significance at the 1% (***), 5% (**), and 10% (*) levels, respectively.

In our second identified regime, however, we find empirical evidence that the equity sectors are impacted by changes in inflation forecasts. Indeed, when inflation is highly scrutinized and forecasts are subject to larger revisions, healthcare and energy show an exposure to implied inflation that is significant at a 1% level with −2.218 and 4.098 coefficients, respectively, while the real estate sector shows a coefficient of 2.152 that is significant at a 5% threshold. Hence, the findings of Amenc et al. (2009) regarding the inflation-hedging properties of real estate and commodities as real assets have to be mitigated for the corresponding equity sectors. However, our findings support the conclusions of Bampinas and Panagiotidis (2016) and Ang et al. (2018), which evidenced the good inflation hedging properties of energy-related stocks. On the one hand, energy and real estate equity sectors are good inflation hedges when investors revise their inflation forecasts substantially, corresponding to periods of reverting realized inflation. However, on the other hand, our findings support that only oil as a real asset benefits from upward anticipated inflation in both regimes. Moreover, among equity sectors, energy stocks are the only ones that are positively exposed to implied inflation and for which the exposure to changes in interest rates is not significant. Consequently, in a regime of large upward revisions of inflation forecasts by market participants with increasing interest rates, energy stocks appear to be the best alternative to inflation-linked securities and oil.

Surprisingly, all interest rate assets show a positive relationship to anticipated inflation in regime 2 when controlling for nominal interest rates. Indeed, bond yields can be decomposed between real yield and anticipated inflation, and, therefore, higher anticipated inflation should lead to higher yields and lower bond prices. However, this effect only exists in one of our two regimes and remains quite low, with changes of 0.4%, 0.6%, and 0.5%, for investment grade bonds, high yield bonds, and convertible bonds, respectively, for a 1% change in implied inflation. The inflation beta of nominal and convertible bonds between 0 and 1 indicates that these assets only offer a partial inflation hedging ability. Thus, our findings support the poor inflation-hedging properties of nominal bonds evidenced by Spierdijk and Umar (2015). As expected, inflation-linked bonds remain strongly correlated with inflation forecasts with a return of 5.9% for a 1% change in implied inflation. As expected too, the sign of the Z-spread coefficient is negative in both regimes for all fixed-income products, consistent with the fact that, all things being equal, bond prices decrease with a higher credit risk.

To ensure the robustness of our results, we have conducted similar regressions on various sub-periods of our sample. The persistence of each regime allows to identify periods featuring both a dominant regime and a sufficient number of observations, so as to obtain meaningful results. More precisely, we have identified four sub-periods, each of them associated with a dominant regime. For each sub-period, we conduct regressions using the data points identified by our regime-switching model as being in the dominant regime. The identified sub-periods are the following: from 19 September 2012 to 12 August 2014 and from 13 January 2017 to 25 February 2019, we consider the dominant regime to be the low uncertainty and low interest about inflation. From 7 October 2014 to 7 December 2016 and from 26 February 2019 to 23 December 2020, we consider the high uncertainty and high interest about inflation to be dominant. For the sake of clarity, results are reported in Appendix A. As one can see, most of the conclusions previously exposed still hold true. Indeed, out of the four sub-periods, oil and inflation-linked securities present a significant inflation beta for three of them. More specifically, for the period from 19 September 2012 to 12 August 2014, none of the assets of our panel show a statistically significant beta to inflation forecasts.[1] Among stock sectors, energy stocks confirm their status of best hedgers with a significant and high inflation beta in both high volatility regimes. On the other hand, real estate stocks, however, do not show empirical evidence of good inflation hedging properties under the high volatility regime between February 2019 and December 2020. However, the Covid-19 crisis has particularly affected real estate stocks in this period and have most likely affected the results for this sub-sample. This brings us to moderate our conclusions on this specific sector: even if there exists evidence of significant

exposure of real estate stocks to implied inflation in periods of high uncertainty about inflation in the long run, their short term inflation-hedging properties can be disappointing. Concerning fixed-income products, our observations are confirmed with stronger betas in the high volatility regime for nominal and inflation-indexed bonds. We also confirm that nominal bonds offer partial inflation hedging ability with inflation betas constantly below 1. Moreover, the weak link between expected inflation and convertible returns seems to be confirmed, with no evidence of a significant inflation beta in any sub-period.

Overall, our results support the hypothesis that inflation-hedging properties of asset classes are regime-dependent Brière and Signori (2012), Spierdijk and Umar (2015). We find that some asset returns can be good hedges when inflation forecasts are significantly revised in the Euro area. More importantly, all asset classes display better inflation-hedging properties in these periods when they are the most needed. However, we confirm that the overall equity market as well as the nominal bond market offer poor hedging properties in the long run in the Euro area, which contradicts the findings of Salisu et al. (2020) in the US stock market. From our perspective, oil-related commodities and inflation-linked government bonds remain the best assets to hedge inflation for a passive investor.

## 6. Conclusions

This paper investigates the non-linear dynamic of market participants' inflation forecasts, and shows that the exposure of European asset returns to implied inflation is regime-dependent. Using a Markov-switching model for European inflation expectations, we identify, historically, a regime of large changes in inflation expectations, corresponding to periods of uncertainty and concerns about realized inflation, and a regime of low volatile inflation forecasts. It appears that the high volatility regime corresponds to periods of reversion for realized volatility.

We have provided empirical evidence that the exposure of European asset returns to inflation forecasts is regime-dependent and strongly sector-dependent. Using a panel of sector equities, bonds, and commodities, we find that oil and inflation-linked bonds are the only assets that offer inflation-hedging properties in both regimes when controlling for a set of macro- and micro-economic variables. However, the positive relationship of inflation-linked bonds in a regime of slow revisions of inflation expectations is small regarding the negative relationship with nominal interest rates. Moreover, alternatives can be found in the stock market for periods of fast and large revisions of inflation forecasts. Indeed, energy and real estate equity sectors appear to be good hedges in a high volatility regime, which is consistent with previous research that demonstrated the good inflation-hedging properties of equities in certain economic regimes, as well as heterogeneous behaviors towards inflation among sectors. Our study confirms that European nominal and convertible bonds present a poor relationship with implied inflation although significant in the high volatility regime on the long term.

As a whole, our findings feature several implications that can be of interest for portfolio managers, investors, and asset-liabilities managers. First, oil can be considered as the best asset to benefit from upward revisions of future inflation, regardless of the regime of inflation expectations. Inflation-linked bonds present good inflation-hedging properties overall, but interest rates movements must be considered since higher interest rates could lead to lower prices for these instruments when market participants slowly adjust their forecasts. During periods of high concerns about inflation, equities in the energy and real estate sectors can be added to the portfolio, although real estate equities suffer from the same negative exposure to interest rates as inflation-linked securities. Finally, bonds offer a partial hedge for inflation expectations in periods of large revisions. Our results support the idea that investors in inflation-linked liabilities such as pension funds should not be passive in their investments. Using a regime-dependent allocation for their liability-hedging portfolio could lead to a better use of their capital.

Finally, our findings tend to indicate that market participants can anticipate structural trend-reversion in realized inflation. Although the quality of their expectations is not

measured, central banks with an inflation target could use implied inflation to quickly assess the credibility of their monetary policy. Future research on the topic includes extending the methodology to larger datasets. Indeed, even if European inflation-linked bonds are still relatively new, other countries such as the United Kingdom and Australia use them for decades. Hence, our study can suffer from side effects such as the ECB's Quantitative Easing. Obtaining similar results for different countries over extended periods would confirm our conclusions in different environments and the robustness of our findings. An alternative solution to get a coherent measure for the breakeven inflation rate in the Euro area would be to compute a complete term structure using the set of ILB issued by Eurozone members using the methodology described in Ejsing et al. (2007). This could provide a few more years of data to analyze. The conclusions would be of particular interest since Spierdijk and Umar (2015) found that inflation-hedging properties of stocks changed dramatically after 2008, a conclusion supported by Bampinas and Panagiotidis (2016). Further research also includes the study of the properties of an optimal inflation-hedging portfolio under regime switches. Indeed, using available information to determine the current regime and transition matrix, one could compute a weighted portfolio of sector-based ETF to get a chosen exposure to inflation expectations. We refer to Ang and Bekaert (2002) for further information about portfolio choice under regime-switching. Our choice of using similar control variables for equities, bonds and commodities can also be a limit to our study. Using asset-specific factors can help to better isolate the role of implied inflation, especially in the case of commodities.

**Author Contributions:** Conceptualization, N.P., J.-P.A., V.J. and F.R.; Data curation, N.P.; Methodology, N.P.; Validation, J.-P.A., V.J. and F.R.; Writing—original draft, N.P.; Writing—review & editing, J.-P.A., V.J. and F.R. All authors have read and agreed to the published version of the manuscript.

**Funding:** This research received no external funding.

**Institutional Review Board Statement:** Not applicable.

**Informed Consent Statement:** Not applicable.

**Data Availability Statement:** Data was downloaded from Bloomberg.

**Conflicts of Interest:** The authors declare no conflict of interest.

## Appendix A. Results of Sub-Periods Regressions

**Table A1.** Results of regressions conducted for our cross-asset panel for the low volatility regime of inflation expectations from 19 September 2012 to 12 August 2014. Error estimates are indicated in parentheses except for the F-Test, for which the *p*-value is in parentheses.

| | Intercept | y | π | ECB | V2X | RM-RF | P/E | P/B | D/P | DIVY | ZSpr | F-Test | Adj. R² |
|---|---|---|---|---|---|---|---|---|---|---|---|---|---|
| Consumer Discretionary | 0.011 (0.021) | −0.184 (0.688) | −0.024 (0.215) | −3.432 ** (1.580) | −0.009 (0.032) | 0.997 *** (0.082) | −0.035 (0.121) | 0.649 (3.295) | −0.020 (0.046) | 1.262 (1.108) | 0.029 (0.021) | 163.81 (0.00) | 0.786 |
| Information Technology | 0.014 (0.030) | 1.449 (0.989) | −0.022 (0.309) | −0.267 (2.273) | 0.025 (0.046) | 0.951 *** (0.117) | −0.107 (0.174) | −0.899 (4.740) | −0.028 (0.066) | 2.350 (1.595) | 0.020 (0.030) | 60.13 (0.00) | 0.571 |
| Industrials | 0.005 (0.016) | −0.581 (0.524) | 0.072 (0.164) | −2.072 * (1.204) | 0.018 (0.024) | 0.932 *** (0.062) | 0.109 (0.092) | 4.013 (2.511) | 0.032 (0.035) | 0.172 (0.845) | 0.006 (0.016) | 300.19 (0.00) | 0.871 |
| Financials | −0.005 (0.022) | 0.802 (0.747) | 0.173 (0.233) | −0.731 (1.717) | −0.024 (0.035) | 1.244 *** (0.089) | −0.044 (0.131) | −2.959 (3.580) | −0.026 (0.050) | −1.818 (1.204) | −0.078 *** (0.023) | 279.50 (0.00) | 0.862 |
| Consumer Staples | 0.017 (0.024) | −0.884 (0.786) | −0.204 (0.245) | 1.588 (1.806) | −0.008 (0.036) | 0.776 *** (0.093) | 0.012 (0.138) | −0.813 (3.767) | 0.007 (0.052) | 1.709 (1.267) | 0.023 (0.024) | 66.80 (0.00) | 0.597 |
| Healthcare | 0.026 (0.029) | 0.267 (0.978) | −0.164 (0.305) | 2.502 (2.247) | −0.028 (0.045) | 0.966 *** (0.116) | 0.044 (0.172) | −3.226 (4.685) | 0.049 (0.065) | 0.633 (1.576) | 0.068 ** (0.030) | 74.80 (0.00) | 0.624 |
| Materials | −0.021 (0.020) | −0.074 (0.677) | −0.020 (0.211) | −1.604 (1.557) | 0.041 (0.031) | 0.910 *** (0.080) | 0.085 (0.119) | 4.609 (3.246) | 0.045 (0.045) | −1.246 (1.092) | 0.024 (0.021) | 181.21 (0.00) | 0.802 |
| Utilities | −0.000 (0.029) | 0.218 (0.973) | 0.173 (0.304) | 2.941 (2.236) | 0.060 (0.045) | 0.958 *** (0.115) | −0.025 (0.171) | −2.905 (4.662) | −0.021 (0.065) | −1.358 (1.568) | −0.013 (0.030) | 75.52 (0.00) | 0.627 |
| Communication Services | −0.021 (0.031) | −0.920 (1.018) | −0.239 (0.318) | 4.276 * (2.340) | 0.011 (0.047) | 1.002 *** (0.121) | −0.122 (0.179) | −3.323 (4.881) | −0.045 (0.068) | −1.135 (1.642) | −0.062 ** (0.031) | 79.44 (0.00) | 0.639 |
| Energy | −0.012 (0.026) | −0.043 (0.856) | 0.019 (0.267) | 3.580 * (1.967) | −0.028 (0.040) | 0.969 *** (0.102) | −0.013 (0.150) | 3.618 (4.102) | 0.010 (0.057) | 1.281 (1.380) | 0.052 ** (0.026) | 112.40 (0.00) | 0.715 |
| Real Estate | −0.015 (0.031) | −4.212 *** (1.042) | 0.443 (0.325) | −0.078 (2.394) | −0.008 (0.048) | 0.744 *** (0.124) | −0.172 (0.183) | 3.867 (4.993) | 0.107 (0.069) | 0.237 (1.679) | −0.062 * (0.032) | 44.51 (0.00) | 0.495 |
| Inv. Grade Bonds | 0.012 *** (0.002) | −3.224 *** (0.077) | −0.005 (0.024) | −0.386 ** (0.178) | −0.002 (0.004) | −0.000 (0.009) | 0.011 (0.014) | 0.230 (0.371) | 0.004 (0.005) | −0.116 (0.125) | −0.033 *** (0.002) | 188.11 (0.00) | 0.808 |
| High Yield Bonds | 0.021 *** (0.003) | −0.957 *** (0.110) | 0.054 (0.034) | −0.246 (0.253) | 0.010 * (0.005) | 0.013 (0.013) | 0.037 * (0.019) | 0.070 (0.527) | 0.010 (0.007) | −0.225 (0.177) | −0.090 *** (0.003) | 97.10 (0.00) | 0.684 |
| Infla. linked Gvt Bonds | 0.005 (0.009) | −3.079 *** (0.300) | 0.056 (0.093) | −0.302 (0.688) | −0.011 (0.014) | 0.060 * (0.036) | −0.010 (0.053) | −0.304 (1.436) | −0.005 (0.020) | 0.055 (0.483) | −0.054 *** (0.009) | 13.32 (0.00) | 0.217 |
| Convertible Bonds | 0.011 (0.007) | −0.100 (0.220) | −0.011 (0.069) | 0.140 (0.505) | −0.003 (0.010) | 0.233 *** (0.026) | 0.004 (0.039) | 0.028 (1.053) | 0.009 (0.015) | 0.333 (0.354) | −0.019 *** (0.007) | 100.31 (0.00) | 0.691 |
| Gold | −0.045 (0.053) | −3.116 * (1.754) | 0.120 (0.547) | 1.742 (4.030) | 0.114 (0.081) | −0.034 (0.208) | 0.096 (0.308) | 13.349 (8.404) | −0.011 (0.117) | 1.359 (2.827) | 0.036 (0.053) | 1.21 (0.28) | 0.005 |
| WTI | −0.021 (0.055) | 2.945 (1.829) | 0.937 (0.571) | 1.186 (4.204) | 0.094 (0.084) | 0.422 * (0.217) | 0.023 (0.321) | 2.131 (8.768) | −0.171 (0.122) | 2.546 (2.949) | 0.096 * (0.056) | 3.12 (0.00) | 0.046 |
| Brent | −0.015 (0.050) | 1.051 (1.676) | 0.731 (0.523) | 4.535 (3.851) | 0.108 (0.077) | 0.393 ** (0.199) | 0.075 (0.294) | 4.104 (8.030) | −0.058 (0.112) | 2.401 (2.701) | 0.126 ** (0.051) | 3.01 (0.00) | 0.043 |

* indicates significance at the 1% (***), 5% (**), and 10% (*) levels, respectively.

**Table A2.** Results of regressions conducted for our cross-asset panel for the high volatility regime of inflation expectations from 7 October 2014 to 7 December 2016. Error estimates are indicated in parentheses except for the F-Test, for which the *p*-value is in parentheses.

| | Intercept | y | π | ECB | V2X | RM-RF | P/E | P/B | D/P | DIVY | ZSpr | F-Test | Adj. R² |
|---|---|---|---|---|---|---|---|---|---|---|---|---|---|
| Consumer Discretionary | 0.024 (0.026) | −0.075 (0.644) | −0.867 (0.832) | −20.220 ** (9.803) | 0.028 (0.026) | 1.072 *** (0.127) | 0.458 * (0.248) | 3.070 (4.843) | −0.131 (0.081) | 4.597 ** (2.017) | −0.007 (0.017) | 310.82 (0.00) | 0.892 |
| Information Technology | −0.004 (0.033) | −1.590 * (0.821) | −2.121 ** (1.060) | 1.067 (12.492) | −0.004 (0.033) | 1.144 *** (0.162) | 0.095 (0.316) | −6.337 (6.172) | 0.011 (0.103) | 3.886 (2.570) | 0.039 * (0.022) | 145.38 (0.00) | 0.793 |
| Industrials | 0.006 (0.018) | −0.831 * (0.459) | 1.286 ** (0.594) | −8.826 (6.994) | 0.009 (0.018) | 0.977 *** (0.090) | −0.171 (0.177) | −4.380 (3.455) | 0.051 (0.057) | −2.612 * (1.439) | 0.012 (0.012) | 515.70 (0.00) | 0.932 |
| Financials | −0.008 (0.033) | 3.056 *** (0.816) | 0.994 (1.055) | 12.144 (12.429) | 0.011 (0.033) | 1.180 *** (0.161) | −0.096 (0.314) | −6.504 (6.141) | −0.025 (0.102) | −2.033 (2.557) | −0.037 * (0.022) | 250.02 (0.00) | 0.869 |
| Consumer Staples | 0.018 (0.031) | −2.594 *** (0.788) | −1.749 * (1.018) | 11.897 (11.999) | 0.022 (0.032) | 0.895 *** (0.155) | −0.159 (0.303) | 6.582 (5.929) | 0.089 (0.099) | 2.048 (2.468) | 0.021 (0.021) | 135.97 (0.00) | 0.782 |
| Healthcare | 0.003 (0.038) | −0.717 (0.956) | −3.257 *** (1.235) | 16.386 (14.547) | −0.034 (0.038) | 0.916 *** (0.188) | 0.312 (0.368) | 8.211 (7.187) | −0.104 (0.120) | 3.391 (2.993) | 0.058 ** (0.025) | 126.81 (0.00) | 0.770 |
| Materials | 0.009 (0.027) | −0.010 (0.682) | 0.846 (0.881) | −3.465 (10.376) | 0.020 (0.027) | 0.922 *** (0.134) | 0.091 (0.262) | 5.189 (5.127) | −0.024 (0.085) | −0.378 (2.135) | 0.010 (0.018) | 267.23 (0.00) | 0.876 |
| Utilities | −0.005 (0.036) | −1.304 (0.903) | −0.383 (1.167) | 7.820 (13.750) | −0.081 ** (0.036) | 0.695 *** (0.178) | −0.493 (0.348) | 2.935 (6.793) | 0.212 * (0.113) | −4.099 (2.829) | −0.022 (0.024) | 112.00 (0.00) | 0.747 |
| Communication Services | −0.006 (0.034) | −0.130 (0.860) | −0.878 (1.111) | −3.861 (13.085) | 0.000 (0.035) | 0.963 *** (0.169) | 0.031 (0.331) | 1.493 (6.465) | −0.039 (0.108) | −0.767 (2.692) | −0.043 * (0.023) | 175.36 (0.00) | 0.823 |
| Energy | −0.026 (0.056) | 0.756 (1.413) | 7.430 *** (1.826) | −21.844 (21.507) | −0.039 (0.057) | 0.893 *** (0.278) | −0.450 (0.544) | −10.033 (10.626) | 0.167 (0.177) | −8.356 * (4.424) | −0.001 (0.037) | 66.36 (0.00) | 0.635 |
| Real Estate | 0.033 (0.044) | −7.790 *** (1.109) | 0.574 (1.433) | −20.996 (16.883) | −0.008 (0.045) | 0.667 *** (0.218) | 0.597 (0.427) | 12.701 (8.341) | −0.201 (0.139) | 4.042 (3.473) | −0.024 (0.029) | 71.11 (0.00) | 0.651 |
| Inv. Grade Bonds | 0.008 ** (0.003) | −3.147 *** (0.087) | 0.206 * (0.112) | −0.566 (1.324) | 0.002 (0.004) | 0.026 (0.017) | 0.006 (0.033) | −0.097 (0.654) | 0.001 (0.011) | 0.441 (0.272) | −0.033 *** (0.002) | 141.85 (0.00) | 0.789 |
| High Yield Bonds | 0.017 *** (0.006) | −1.077 *** (0.158) | 0.567 *** (0.204) | 0.630 (2.408) | −0.011 * (0.006) | 0.052 * (0.031) | 0.050 (0.061) | −0.799 (1.190) | −0.020 (0.020) | 0.398 (0.495) | −0.085 *** (0.004) | 86.61 (0.00) | 0.695 |
| Infla. linked Gvt Bonds | 0.009 (0.010) | −5.374 *** (0.250) | 4.556 *** (0.323) | −2.545 (3.809) | −0.011 (0.010) | 0.056 (0.049) | 0.090 (0.096) | 1.677 (1.882) | −0.031 (0.031) | 1.238 (0.783) | −0.020 *** (0.007) | 66.10 (0.00) | 0.634 |
| Convertible Bonds | 0.002 (0.007) | −0.799 *** (0.173) | −0.181 (0.224) | −2.557 (2.637) | 0.001 (0.007) | 0.232 *** (0.034) | 0.150 ** (0.067) | 3.770 *** (1.303) | −0.044 ** (0.022) | 0.551 (0.543) | −0.015 *** (0.005) | 373.86 (0.00) | 0.908 |
| Gold | 0.002 (0.050) | −1.429 (1.242) | 0.504 (1.605) | −48.016 ** (18.911) | −0.033 (0.050) | −0.021 (0.245) | 0.262 (0.478) | −0.286 (9.344) | 0.001 (0.155) | 6.062 (3.890) | 0.025 (0.033) | 3.27 (0.00) | 0.057 |
| WTI | −0.149 (0.155) | 6.424 * (3.881) | 23.506 *** (5.014) | −120.242 ** (59.078) | 0.104 (0.156) | 0.434 (0.764) | −0.931 (1.494) | −26.639 (29.189) | 0.256 (0.486) | −17.577 (12.153) | −0.000 (0.103) | 5.93 (0.00) | 0.116 |
| Brent | −0.102 (0.145) | 6.049 * (3.641) | 23.450 *** (4.704) | −85.021 (55.420) | 0.077 (0.147) | 0.209 (0.717) | −1.049 (1.402) | −19.245 (27.382) | 0.275 (0.456) | −20.950 * (11.401) | −0.013 (0.096) | 6.61 (0.00) | 0.130 |

* indicates significance at the 1% (***), 5% (**), and 10% (*) levels, respectively.

**Table A3.** Results of regressions conducted for our cross-asset panel for the low volatility regime of inflation expectations from 13 January 2017 to 25 February 2019. Error estimates are indicated in parentheses except for the F-Test, for which the *p*-value is in parentheses.

| | Intercept | y | π | ECB | V2X | RM-RF | P/E | P/B | D/P | DIVY | ZSpr | F-Test | Adj. R² |
|---|---|---|---|---|---|---|---|---|---|---|---|---|---|
| Consumer Discretionary | −0.001 (0.018) | −1.960*** (0.745) | −0.185 (1.603) | −0.000 (0.000) | 0.035* (0.019) | 1.263*** (0.111) | −0.141 (0.137) | −0.678 (5.157) | 0.049 (0.065) | 1.565 (2.107) | −0.000 (0.019) | 224.72 (0.00) | 0.790 |
| Information Technology | 0.022 (0.028) | −4.945*** (1.137) | −0.903 (2.444) | −0.000 (0.000) | 0.017 (0.030) | 1.349*** (0.169) | −0.025 (0.208) | −2.330 (7.863) | −0.070 (0.100) | 2.740 (3.212) | −0.005 (0.029) | 105.68 (0.00) | 0.638 |
| Industrials | 0.012 (0.013) | −2.500*** (0.531) | −0.483 (1.141) | 0.000 (0.000) | 0.024* (0.014) | 1.007*** (0.079) | 0.029 (0.097) | 4.318 (3.672) | −0.028 (0.046) | −0.897 (1.500) | 0.008 (0.014) | 395.19 (0.00) | 0.869 |
| Financials | −0.011 (0.020) | 10.149*** (0.837) | 0.746 (1.800) | 0.000 (0.000) | 0.037* (0.022) | 0.872*** (0.124) | 0.107 (0.153) | 2.965 (5.789) | −0.022 (0.073) | −3.661 (2.365) | −0.051** (0.022) | 219.46 (0.00) | 0.786 |
| Consumer Staples | 0.009 (0.022) | −4.373*** (0.898) | −3.122 (1.931) | −0.000 (0.000) | −0.054** (0.023) | 0.728*** (0.133) | −0.050 (0.165) | −0.648 (6.213) | 0.046 (0.079) | 1.470 (2.538) | 0.038 (0.023) | 62.15 (0.00) | 0.508 |
| Healthcare | −0.034 (0.023) | −2.038** (0.926) | −1.577 (1.991) | −0.000** (0.000) | −0.017 (0.024) | 1.210*** (0.138) | 0.042 (0.170) | −12.247* (6.407) | −0.016 (0.081) | 1.823 (2.617) | 0.024 (0.024) | 108.17 (0.00) | 0.644 |
| Materials | −0.005 (0.018) | −0.632 (0.749) | 1.537 (1.611) | 0.000 (0.000) | 0.033* (0.020) | 1.141*** (0.111) | −0.109 (0.137) | 6.209 (5.183) | 0.067 (0.066) | −0.520 (2.117) | 0.015 (0.019) | 252.22 (0.00) | 0.809 |
| Utilities | 0.046 (0.028) | −3.822*** (1.144) | −0.656 (2.459) | −0.000 (0.000) | −0.094*** (0.030) | 0.671*** (0.170) | 0.056 (0.210) | −3.487 (7.912) | −0.020 (0.100) | −1.214 (3.232) | 0.003 (0.030) | 48.74 (0.00) | 0.446 |
| Communication Services | −0.018 (0.026) | 0.378 (1.052) | 0.590 (2.262) | 0.000 (0.000) | −0.092*** (0.027) | 0.615*** (0.156) | −0.007 (0.193) | 4.176 (7.277) | 0.037 (0.092) | 0.822 (2.972) | 0.012 (0.027) | 60.02 (0.00) | 0.499 |
| Energy | 0.018 (0.028) | −0.123 (1.129) | 8.645*** (2.428) | 0.000 (0.000) | −0.026 (0.029) | 0.906*** (0.168) | 0.124 (0.207) | −5.479 (7.810) | −0.048 (0.099) | 0.272 (3.190) | 0.019 (0.029) | 64.22 (0.00) | 0.516 |
| Real Estate | −0.018 (0.030) | −6.226*** (1.230) | −1.003 (2.644) | −0.000 (0.000) | −0.125*** (0.032) | 0.687*** (0.183) | −0.097 (0.225) | −7.528 (8.506) | −0.016 (0.108) | 4.160 (3.475) | −0.024 (0.032) | 26.52 (0.00) | 0.301 |
| Inv. Grade Bonds | 0.006*** (0.002) | −3.883*** (0.075) | 0.222 (0.161) | 0.000 (0.000) | 0.001 (0.002) | 0.004 (0.011) | 0.000 (0.014) | 0.095 (0.519) | 0.002 (0.007) | 0.060 (0.212) | −0.040*** (0.002) | 330.44 (0.00) | 0.847 |
| High Yield Bonds | 0.012*** (0.003) | −1.431*** (0.122) | 0.540** (0.262) | −0.000 (0.000) | −0.001 (0.003) | 0.042** (0.018) | 0.006 (0.022) | −1.285 (0.843) | −0.005 (0.011) | −0.497 (0.344) | −0.087*** (0.003) | 143.64 (0.00) | 0.706 |
| Infla. linked Gvt Bonds | 0.008 (0.008) | −5.035*** (0.320) | 4.857*** (0.689) | 0.000*** (0.000) | −0.007 (0.008) | −0.104** (0.048) | −0.101* (0.059) | 5.910*** (2.216) | 0.063** (0.028) | −2.042** (0.905) | −0.056*** (0.008) | 33.35 (0.00) | 0.353 |
| Convertible Bonds | 0.002 (0.005) | −1.437*** (0.216) | 0.224 (0.465) | 0.000 (0.000) | −0.007 (0.006) | 0.316*** (0.032) | 0.015 (0.040) | 0.076 (1.496) | −0.023 (0.019) | 0.824 (0.611) | −0.011** (0.006) | 199.19 (0.00) | 0.770 |
| Gold | 0.017 (0.027) | −3.994*** (1.093) | 0.173 (2.350) | −0.000* (0.000) | −0.059** (0.028) | 0.030 (0.162) | 0.118 (0.200) | −15.021** (7.562) | −0.084 (0.096) | 0.333 (3.089) | −0.044 (0.028) | 3.72 (0.00) | 0.044 |
| WTI | 0.024 (0.073) | −7.208** (2.978) | 30.932*** (6.404) | 0.000* (0.000) | −0.027 (0.078) | 0.015 (0.442) | −0.230 (0.546) | 8.093 (20.603) | −0.068 (0.261) | −12.094 (8.416) | 0.074 (0.077) | 6.30 (0.00) | 0.082 |
| Brent | 0.037 (0.070) | −6.267** (2.871) | 29.610*** (6.175) | 0.000 (0.000) | −0.037 (0.075) | 0.331 (0.426) | −0.353 (0.526) | −8.004 (19.864) | 0.053 (0.251) | −10.815 (8.114) | 0.065 (0.074) | 6.60 (0.00) | 0.086 |

\* indicates significance at the 1% (\*\*\*), 5% (\*\*), and 10% (\*) levels, respectively.

**Table A4.** Results of regressions conducted for our cross-asset panel for the high volatility regime of inflation expectations from 26 February 2019 to 23 December 2020. Error estimates are indicated in parentheses except for the F-Test, for which the *p*-value is in parentheses.

| | Intercept | y | π | ECB | V2X | RM-RF | P/E | P/B | D/P | DIVY | ZSpr | F-Test | Adj. R² |
|---|---|---|---|---|---|---|---|---|---|---|---|---|---|
| Consumer Discretionary | 0.047 * (0.028) | 0.980 (1.086) | 0.275 (1.435) | −0.000 (0.000) | 0.029 (0.021) | 1.322 *** (0.118) | −0.298 *** (0.097) | −6.648 (6.173) | 0.165 *** (0.054) | 1.297 (1.107) | 0.022 * (0.012) | 294.72 (0.00) | 0.889 |
| Information Technology | 0.022 (0.048) | −2.901 (1.880) | 2.243 (2.484) | 0.000 (0.000) | −0.091 ** (0.037) | 0.595 *** (0.204) | 0.220 (0.168) | 15.530 (10.684) | −0.121 (0.094) | −1.512 (1.915) | 0.018 (0.021) | 92.08 (0.00) | 0.713 |
| Industrials | −0.006 (0.025) | −0.755 (0.958) | −0.277 (1.265) | −0.000 (0.000) | 0.121 *** (0.019) | 1.411 *** (0.104) | −0.066 (0.085) | −6.188 (5.442) | 0.044 (0.048) | 0.133 (0.976) | −0.003 (0.011) | 450.25 (0.00) | 0.925 |
| Financials | −0.007 (0.044) | 10.022 *** (1.718) | −1.241 (2.269) | 0.000 * (0.000) | 0.158 *** (0.034) | 1.239 *** (0.186) | 0.153 (0.153) | 12.094 (9.761) | −0.087 (0.086) | 2.540 (1.750) | −0.026 (0.019) | 192.15 (0.00) | 0.839 |
| Consumer Staples | −0.002 (0.035) | −4.168 *** (1.350) | 0.388 (1.784) | −0.000 (0.000) | −0.056 ** (0.027) | 0.685 *** (0.146) | 0.093 (0.121) | −0.486 (7.672) | −0.056 (0.067) | −0.163 (1.375) | −0.010 (0.015) | 97.89 (0.00) | 0.725 |
| Healthcare | −0.028 (0.043) | −4.472 *** (1.675) | −1.592 (2.213) | 0.000 (0.000) | −0.145 *** (0.033) | 0.448 ** (0.181) | 0.187 (0.150) | 1.844 (9.518) | −0.115 (0.084) | −3.214 * (1.706) | 0.010 (0.018) | 64.96 (0.00) | 0.636 |
| Materials | 0.031 (0.028) | 1.464 (1.093) | −1.324 (1.443) | −0.000 ** (0.000) | −0.013 (0.022) | 1.408 *** (0.118) | −0.283 *** (0.098) | −18.002 *** (6.209) | 0.163 *** (0.055) | 1.493 (1.113) | 0.009 (0.012) | 278.30 (0.00) | 0.883 |
| Utilities | 0.005 (0.040) | −8.129 *** (1.560) | 0.241 (2.061) | −0.000 ** (0.000) | −0.090 *** (0.031) | 0.922 *** (0.169) | 0.007 (0.139) | −14.567 (8.863) | 0.015 (0.078) | −0.693 (1.589) | −0.016 (0.017) | 82.07 (0.00) | 0.689 |
| Communication Services | −0.020 (0.039) | 0.116 (1.531) | −2.230 (2.022) | −0.000 (0.000) | −0.112 *** (0.030) | 0.858 *** (0.166) | −0.209 (0.137) | −11.039 (8.699) | 0.120 (0.076) | −1.018 (1.559) | 0.034 ** (0.017) | 78.37 (0.00) | 0.678 |
| Energy | −0.050 (0.067) | 3.692 (2.605) | 5.988 * (3.441) | 0.000 (0.000) | 0.009 (0.051) | 0.608 ** (0.282) | 0.427 * (0.232) | 32.996 ** (14.801) | −0.281 ** (0.130) | −0.096 (2.653) | −0.016 (0.029) | 81.69 (0.00) | 0.688 |
| Real Estate | −0.087 (0.058) | −4.481 ** (2.262) | −1.299 (2.987) | −0.000 (0.000) | 0.078 * (0.045) | 0.768 *** (0.245) | 0.031 (0.202) | −8.002 (12.850) | 0.028 (0.113) | −4.281 * (2.304) | −0.037 (0.025) | 31.51 (0.00) | 0.454 |
| Inv. Grade Bonds | 0.006 ** (0.002) | −4.274 *** (0.092) | 0.245 ** (0.122) | −0.000 *** (0.000) | 0.003 (0.002) | 0.039 *** (0.010) | −0.002 (0.008) | −2.297 *** (0.524) | 0.004 (0.005) | −0.023 (0.094) | −0.050 *** (0.001) | 501.21 (0.00) | 0.932 |
| High Yield Bonds | 0.008 (0.007) | −1.636 *** (0.269) | 0.832 ** (0.356) | −0.000 * (0.000) | 0.007 (0.005) | 0.035 (0.029) | 0.015 (0.024) | −0.728 (1.531) | −0.008 (0.013) | −0.215 (0.274) | −0.106 *** (0.003) | 279.29 (0.00) | 0.884 |
| Infla. linked Gvt Bonds | 0.002 (0.010) | −7.892 *** (0.397) | 6.653 *** (0.524) | −0.000 *** (0.000) | −0.004 (0.008) | 0.023 (0.043) | −0.013 (0.035) | 0.576 (2.256) | 0.007 (0.020) | −0.372 (0.404) | −0.031 *** (0.004) | 60.69 (0.00) | 0.619 |
| Convertible Bonds | 0.015 (0.011) | −2.366 *** (0.428) | 0.727 (0.566) | −0.000 (0.000) | −0.033 *** (0.008) | 0.115 ** (0.046) | 0.079 ** (0.038) | 0.173 (2.433) | −0.042 ** (0.021) | −0.513 (0.436) | −0.015 *** (0.005) | 72.87 (0.00) | 0.662 |
| Gold | 0.063 (0.055) | −8.063 *** (2.124) | 1.101 (2.805) | −0.000 (0.000) | −0.219 *** (0.042) | −0.095 (0.230) | −0.515 *** (0.190) | −0.119 (12.067) | 0.264 ** (0.106) | 2.666 (2.163) | −0.023 (0.023) | 6.47 (0.00) | 0.130 |
| WTI | −1.434 (1.049) | −68.530 * (40.717) | 101.429 * (53.781) | −0.000 (0.000) | −1.260 (0.802) | 0.154 (4.407) | 0.315 (3.634) | −108.079 (231.342) | −0.084 (2.031) | −21.239 (41.473) | −0.234 (0.447) | 1.17 (0.31) | 0.005 |
| Brent | −0.131 (0.165) | −9.514 (6.394) | 26.296 *** (8.446) | −0.000 (0.000) | −0.201 (0.126) | 0.434 (0.692) | −0.452 (0.571) | −4.545 (36.330) | 0.200 (0.319) | −9.764 (6.513) | −0.111 (0.070) | 11.59 (0.00) | 0.224 |

* indicates significance at the 1% (***), 5% (**), and 10% (*) levels, respectively.

## Note

1    Although this period corresponds to a regime of low interest for inflation from market participants, it would be of interest to find out the fundamental causes behind this result.

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
