# Peer review of "Inflation Forecasts and European Asset Returns: A Regime-Switching Approach"

_jrfm, doi:10.3390/jrfm15100475_

Round 1

Reviewer 1 Report

JRFM: Inflation Forecasts and European Asset Returns : A Regime-Switching Approach

I like the idea of the paper and its execution, especially in light of the growing interested in how asset return are affects by the rising level of inflation after the supply chain interruption under covid and the rise in commodity prices during the war in Ukraine.

However, there are some issues that should be addressed regarding the literature review and the added value of the results. Once addressed, I will reconsider my decision. My main comments are as follows:

1-    The introduction section:

2-    The literature review: includes papers published in the past decades, where no recent papers are used to position the paper within the related literature on hedging. Therefore, I suggest to relate the paper to a recent strand of literature on the other assets that can be used to hedge risky assets such as equity. The other assets include gold, crude oil, commodity, and Bitcoin. Then, the authors can say that unlike this strand of literature, we are focusing on the hedging ability of equity for inflation risk. Thorough this, the authors can serve two objectives, 1) better position their analysis to the board hedge literature, and 2) cite recent papers published after 2019. In this regard, I suggest the following papers: (i) Revisiting the valuable roles of commodities for international stock markets. Resources Policy, Vol. 66, 101603. (ii) The dynamic network connectedness and hedging strategies across stock markets and commodities: COVID-19 pandemic effect. Asia-Pacific Journal of Business Administration. Vol. 13 No. 4, pp. 520-552 (iii) Safe haven, hedge and diversification for G7 stock markets: Gold versus bitcoin. Economic Modelling, Vol. 87, pp. 212-224.(iv) Is Bitcoin a better safe-haven investment than gold and commodities? International Review of Financial Analysis, Vol. 63, pp. 322-330. 

3-    The data and methods: The choice of the indices and variables need further motivation, notably the choice of the proxy of implied volatility in the EU. Furthermore, I wonder why the authors opt for the parametric and non-parametric non-linearity tests? Please justify further the choice.

4-    The results: I wonder why applying the MS to the AR term? Why not only the inflation expectations? Furthermore, try to better explain the results in light of previous studies to underline the added value of your analysis.

5-    The policy implications: Consider a more detailed discussion of the policy implications and study limitations and scope for future research.

6-    Tables and figures: make your tables more self-explanatory.

7-    References: Make sure that all cited papers are presented in the reference list and vice-versa.

Reviewer 2 Report

Overall the manuscript is well written, and the results are soundly presented. The manuscript is required to address the following comments before its publication JFRM.

1.     There is no segregation of introduction and literature. I find significantly less number of current studies in the manuscript.

2.     Its not clear why the period from 2009 to 2021 is taken in the current study.

3.     There is a need to expand the conclusion and add a discussion of results to defend or support the results obtained by the study.

4.     Implications or contribution of the study is missing to policymakers and other stakeholders. The authors are suggested to include academic implications and further scope of research.

5.     There is an exhausitive list of control variables  (Table 2)

6.     Hypotheses in the study are missing. Its recommended to include hypotheses.

7.     It is stated in the study that "Our objective is to identify the statistical significance of both control variables and implied inflation for each asset category of our cross-asset panel." It's good that implied inflation should be significant. But why the control variables are also intended to be significant?

8.     Authors must include diagnostic statistics and tools applied to check the robustness of the models developed in the current study.

Reviewer 3 Report

Your paper is well-prepared, clearly structured & presented in a logical manner. It has a straight-forward analysis which in turn, is appropriate for this type of study. The practical implications are well presented in the last paragraph along with limitations and future research recommendations. 

Two minor things are recommended: 

- Some minor grammatical errors are identified, therefore, the paper needs a soft proofreading. 

- A short sentence about significance and value-adding of the study is need in the introduction section.  

Round 2

Reviewer 1 Report

I accept this revised manuscript.